# Softstar: Heuristic-Guided Probabilistic Inference

**Mathew Monfort**
Computer Science Department
University of Illinois at Chicago
Chicago, IL 60607
mmonfo2@uic.edu

**Brenden M. Lake**
Center for Data Science
New York University
New York, NY 10003
brenden@nyu.edu

**Brian D. Ziebart**
Computer Science Department
University of Illinois at Chicago
Chicago, IL 60607
bziebart@uic.edu

**Patrick Lucey**
Disney Research Pittsburgh
Pittsburgh, PA 15232
patrick.lucey@disneyresearch.com

**Joshua B. Tenenbaum**
Brain and Cognitive Sciences Department
Massachusetts Institute of Technology
Cambridge, MA 02139
jbt@mit.edu

## Abstract

Recent machine learning methods for sequential behavior prediction estimate the motives of behavior rather than the behavior itself. This higher-level abstraction improves generalization in different prediction settings, but computing predictions often becomes intractable in large decision spaces. We propose the *Softstar* algorithm, a softened heuristic-guided search technique for the maximum entropy inverse optimal control model of sequential behavior. This approach supports probabilistic search with bounded approximation error at a significantly reduced computational cost when compared to sampling based methods. We present the algorithm, analyze approximation guarantees, and compare performance with simulation-based inference on two distinct complex decision tasks.

## 1 Introduction

Inverse optimal control (IOC) [13], also known as inverse reinforcement learning [18, 1] and inverse planning [3], has become a powerful technique for learning to control or make decisions based on expert demonstrations [1, 20]. IOC estimates the utilities of a decision process that rationalizes an expert's demonstrated control sequences. Those estimated utilities can then be used in an (optimal) controller to solve new decision problems, producing behavior that is similar to demonstrations.

Predictive extensions to IOC [17, 23, 2, 16, 19, 6] recognize the inconsistencies, and inherent sub-optimality, of repeated behavior by incorporating uncertainty. They provide probabilistic forecasts of future decisions in which stochasticity is due to this uncertainty rather than the stochasticity of the decision process's dynamics. These models' distributions over plans and policies can typically be defined as softened versions of optimal sequential decision criteria.

A key challenge for predictive IOC is that many decision sequences are embedded within large decision processes. Symmetries in the decision process can be exploited to improve efficiency [21], but decision processes are not guaranteed to be (close to) symmetric. Approximation approaches to probabilistic structured prediction include approximate maxent IOC [12], heuristic-guided sampling [15], and graph-based IOC [7]. However, few guarantees are provided by these approaches; they are not complete and the set of variable assignments uncovered may not be representative of the model's distribution.

Seeking to provide stronger guarantees and improve efficiency over previous methods, we present *Softstar*, a heuristic-guided probabilistic search algorithm for inverse optimal control. Our approach

generalizes the A* search algorithm [8] to calculate distributions over decision sequences in predictive IOC settings allowing for efficient bounded approximations of the near-optimal path distribution through a decision space. This distribution can then be used to update a set of trainable parameters, $\theta$, that motivate the behavior of the decision process via a cost/reward function [18, 1, 3, 23].

We establish theoretical guarantees of this approach and demonstrate its effectiveness in two settings: learning stroke trajectories for Latin characters and modeling the ball-handling decision process of professional soccer.

## 2 Background

### 2.1 State-space graphs and Heuristic-guided search

In this work, we restrict our consideration to deterministic planning tasks with discrete state spaces. The space of plans and their costs can be succinctly represented using a state-space graph, $\mathcal{G} = (\mathcal{S}, \mathcal{E}, cost)$. With vertices, $s \in \mathcal{S}$, representing states of the planning task and directed edges, $e_{ab} \in \mathcal{E}$, representing available transitions between states $s_a$ and $s_b$. The neighbor set of state $s$, $\mathcal{N}(s)$, is the set of states to which $s$ has a directed edge and a cost function, $cost(s, s')$, represents the relative desirability of transitioning between states $s$ and $s'$.

The optimal plan from state $s_1$ to goal state $s_T$ is a variable-length sequence of states $(s_1, s_2, \ldots, s_T)$ forming a path through the graph minimizing a cumulative penalty. Letting $h(s)$ represent the cost of the optimal path from state $s$ to state $s_T$ (i.e., the cost-to-go or value of $s$) and defining $h(s_T) \triangleq 0$, the optimal path corresponds to a fixed-point solution of the next state selection criterion [5]:

$$h(s) = \min_{s' \in \mathcal{N}(s)} h(s') + cost(s, s'), \qquad s_{t+1} = \operatorname*{argmin}_{s' \in \mathcal{N}(s_t)} h(s') + cost(s_t, s'). \qquad (1)$$

The optimal path distance to the start state, $d(s)$, can be similarly defined (with $d(s_1) \triangleq 0$) as

$$d(s) = \min_{s': s \in N(s')} d(s') + cost(s', s). \qquad (2)$$

Dynamic programming algorithms, such as Dijkstra's [9], search the space of paths through the state-space graph in order of increasing $d(s)$ to find the optimal path. Doing so implicitly considers all paths up to the length of the optimal path to the goal.

Additional knowledge can significantly reduce the portion of the state space needed to be explored to obtain an optimal plan. For example, A* search [11] explores partial state sequences by expanding states that minimize an estimate, $f(s) = d(s) + \hat{h}(s)$, combining the minimal cost to reach state $s$, $d(s)$, with a heuristic estimate of the remaining cost-to-go, $\hat{h}(s)$. A priority queue is used to keep track of expanded states and their respective estimates. A* search then expands the state at the top of the queue (lowest $f(s)$) and adds its neighboring states to the queue. When the heuristic estimate is admissible (i.e. $\hat{h}(s) \leq h(s) \ \forall \ s \in \mathcal{S}$), the algorithm terminates with a guaranteed optimal solution once the best "unexpanded" state's estimate, $f(s)$, is worse than the best discovered path to the goal.

### 2.2 Predictive inverse optimal control

Maximum entropy IOC algorithms [23, 22] estimate a stochastic action policy that is most uncertain while still guaranteeing the same expected cost as demonstrated behavior on an unknown cost function [1]. For planning settings with deterministic dynamics, this yields a probability distribution over state sequences that are consistent with paths through the state-space graph, $\hat{P}(\mathbf{s}_{1:T}) \propto e^{-\text{cost}_\theta(\mathbf{s}_{1:T})}$, where $\text{cost}_\theta(\mathbf{s}_{1:T}) = \sum_{t=1}^{T-1} \theta^{\mathrm{T}} \mathbf{f}(s_t, s_{t+1})$ is a linearly weighted vector of state-transition features combined using the feature function, $\mathbf{f}(\mathbf{s}_t, \mathbf{s}_{t+1})$, and a learned parameter vector, $\theta$. Calculating the marginal state probabilities of this distribution is important for estimating model parameters. The forward-backward algorithm [4] can be employed, but for large state-spaces it may not be practical.

# 3 Approach

Motivated by the efficiency of heuristic-guided search algorithms for optimal planning, we define an analogous approximation task in the predictive inference setting and present an algorithm that leverages heuristic functions to accomplish this task efficiently with bounded approximation error.

The problem being addressed is the inefficiency of existing inference methods for reward/cost-based probabilistic models of behavior. We present a method using ideas from heuristic-guided search (i.e., A*) for estimating path distributions through large scale deterministic graphs with bounded approximation guarantees. This is an improvement over previous methods as it results in more accurate distribution estimations without the complexity/sub-optimality concerns of path sampling and is suitable for any problem that can be represented as such a graph.

Additionally, since the proposed method does not sample paths, but instead searches the space as in A*, it does not need to retrace its steps along a previously searched trajectory to find a new path to the goal. It will instead create a new branch from an already explored state. Sampling would require retracing an entire sequence until this branching state was reached. This allows for improvements in efficiency in addition to the distribution estimation improvements.

## 3.1 Inference as softened planning

We begin our investigation by recasting the inference task from the perspective of softened planning where the predictive IOC distribution over state sequences factors into a stochastic policy [23],

$$\pi(s_{t+1}|s_t) = e^{h_{\text{soft}}(s_t) - h_{\text{soft}}(s_{t+1}) - \theta^{\text{T}}\mathbf{f}(s_t, s_{t+1})}, \tag{3}$$

according to a softened cost-to-go , $h_{\text{soft}}(s)$, recurrence that is a relaxation of the Bellman equation:

$$h_{\text{soft}}(s_t) = -\log \sum_{s_{t:T} \in \Xi_{s_t, s_{\text{T}}}} e^{-\text{cost}_\theta(\mathbf{s}_{t:T})} = \underset{s_{t+1} \in \mathcal{N}(s_t)}{\text{softmin}} \left\{ h_{\text{soft}}(s_{t+1}) + \theta^{\text{T}}\mathbf{f}(s_t, s_{t+1}) \right\} \tag{4}$$

where $\Xi_{s_t, s_{\text{T}}}$ is the set of all paths from $s_t$ to $s_{\text{T}}$; the softmin, $\underset{x}{\text{softmin}}\, \alpha(x) \triangleq -\log \sum_x e^{-\alpha(x)}$, is a smoothed relaxation of the $\min$ function[1], and the goal state value is initially $0$ and $\infty$ for others.

A similar softened minimum distance exists in the forward direction from the start state,

$$d_{\text{soft}}(s_t) = -\log \sum_{s_{1:t} \in \Xi_{s_1, s_t}} e^{-\text{cost}_\theta(\mathbf{s}_{1:t})} = \underset{s_{t-1} \in \mathcal{N}(s_t)}{\text{softmin}} \left\{ d_{\text{soft}}(s_{t-1}) + \theta^{\text{T}}\mathbf{f}(s_{t-1}, s_t) \right\}.$$

By combining forward and backward soft distances, important marginal expectations are obtained and used to predict state visitation probabilities and fit the maximum entropy IOC model's parameters [23]. Efficient search and learning require accurate estimates of $d_{\text{soft}}$ and $h_{\text{soft}}$ values since the expected number of occurrences of the transition from $s_a$ to $s_b$ under the soft path distribution is:

$$e^{-d_{\text{soft}}(s_a) - h_{\text{soft}}(s_b) - \theta^{\text{T}}\mathbf{f}(s_a, s_b) + d_{\text{soft}}(s_{\text{T}})}. \tag{5}$$

These cost-to-go and distance functions can be computed in closed-form using a geometric series,

$$\mathbf{B} = \mathbf{A}(\mathbf{I} - \mathbf{A})^{-1} = \mathbf{A} + \mathbf{A}^2 + \mathbf{A}^3 + \mathbf{A}^4 + \cdots, \tag{6}$$

where $A_{i,j} = e^{-cost(s_i, s_j)}$ for any states $s_i$ and $s_j \in \mathcal{S}$.

The $(i, j)^{th}$ entry of $\mathbf{B}$ is related to the softmin of all the paths from $s_i$ to $s_j$. Specifically, the softened cost-to-go can be written as $h_{soft}(s_i) = -\log b_{s_i, s_{\text{T}}}$. Unfortunately, the required matrix inversion operation is computationally expensive, preventing its use in typical inverse optimal control applications. In fact, power iteration methods used for sparse matrix inversion closely resemble the softened Bellman updates of Equation (4) that have instead been used for IOC [22].

## 3.2 Challenges and approximation desiderata

In contrast with optimal control and planning tasks, softened distance functions, $d_{\text{soft}}(s)$, and cost-to-go functions, $h_{\text{soft}}(s)$, in predictive IOC are based on many paths rather than a single (best) one. Thus, unlike in A* search, each sub-optimal path cannot simply be ignored; its influence must instead be incorporated into the softened distance calculation (4). This key distinction poses a significantly different objective for heuristic-guided probabilistic search: *Find a subset of paths for which the softmin distances closely approximate the softmin of the entire path set.* While we would hope that a small subset of paths exists that provides a close approximation, the cost function weights and the structure of the state-space graph ultimately determine if this is the case. With this in mind, we aim to construct a method with the following desiderata for an algorithm that seeks a small approximation set and analyze its guarantees:

1. *Known bounds on approximation guarantees;*
2. *Convergence to any desired approximation guarantee;*
3. *Efficienct finding small approximation sets of paths.*

## 3.3 Regimes of Convergence

In A* search, theoretical results are based on the assumption that all infinite length paths have infinite cost (i.e., any cycle has a positive cost) [11]. This avoids a negative cost cycle regime of non-convergence. Leading to a stronger requirement for our predictive setting are three regimes of convergence for the predictive IOC distribution, characterized by:

1. *An infinite-length most likely plan;*
2. *A finite-length most likely plan with expected infinite-length plans; and*
3. *A finite expected plan length.*

The first regime results from the same situation described for optimal planning: reachable cycles of negative cost. The second regime arises when the number of paths grows faster than the penalization of the weights from the additional cost of longer paths (without negative cycles) and is non-convergent. The final regime is convergent.

An additional assumption is needed in the predictive IOC setting to avoid the second regime of non-convergence. We assume that a fixed bound on the entropy of the distribution of paths, $H(\mathbf{S}_{1:T}) \triangleq \mathbb{E}[-\log P(\mathbf{S}_{1:T})] \leq H_{\max}$, is known.

**Theorem 1** *Expected costs under the predictive IOC distribution are related to entropy and softmin path costs by* $\mathbb{E}[cost_\theta(\mathbf{S}_{1:T})] = H(\mathbf{S}_{1:T}) - d_{soft}(s_T)$.

Together, bounds on the entropy and softmin distance function constrain expected costs under the predictive IOC distribution (Theorem 1).

## 3.4 Computing approximation error bounds

A* search with a non-monotonic heuristic function guarantees optimality when the priority queue's minimal element has an estimate $d_{\text{soft}}(s) + \hat{h}_{\text{soft}}(s)$ exceeding the best start-to-goal path cost, $d_{\text{soft}}(s_\text{T})$. Though optimality is no longer guaranteed in the softmin search setting, approximations to the softmin distance are obtained by considering a subset of paths (Lemma 1).

**Lemma 1** *Let $\Xi$ represent the entire set (potentially infinite in size) of paths from state $s$ to $s_T$. We can partition the set $\Xi$ into two sets $\Xi_a$ and $\Xi_b$ such that $\Xi_a \cup \Xi_b = \Xi$ and $\Xi_a \cap \Xi_b = \emptyset$ and define $d_{soft}^{\Xi}$ as the softmin over all paths in set $\Xi$. Then, given a lower bound estimate for the distance, $\hat{d}_{soft}(s) \leq d_{soft}(s)$, we have $e^{-d_{soft}^{\Xi}(s)} - e^{-d_{soft}^{\Xi_a}(s)} \leq e^{-\hat{d}_{soft}^{\Xi_b}(s)}$.*

We establish a bound on the error introduced by considering the set of paths through a set of states $\mathcal{S}_\approx$ in the following Theorem.

**Theorem 2** *Given an approximation state subset $\mathcal{S}_\approx \subseteq \mathcal{S}$ with neighbors of the approximation set defined as $\mathcal{N}(\mathcal{S}_\approx) \triangleq \bigcup_{s \in \mathcal{S}_\approx} \mathcal{N}(s) - \mathcal{S}_\approx$, the approximation loss of exact search for paths through*

*this approximation set (i.e., paths with non-terminal vertices from $\mathcal{S}_{\approx}$ and terminal vertices from $\mathcal{S}_{\approx} \cup \mathcal{N}(\mathcal{S}_{\approx})$) is bounded by the softmin of the set's neighbors estimates, $e^{-d_{soft}(s_T)} - e^{-d_{soft}^{\mathcal{S}_{\approx}}(s_T)} \leq e^{-\operatorname{softmin}_{s \in \mathcal{N}(\mathcal{S}_{\approx})}\left\{d_{soft}^{\mathcal{S}_{\approx}}(s) + \hat{h}_{soft}(s)\right\}}$, where $d_{soft}^{\mathcal{S}_{\approx}}(s)$ is the softmin of all paths with terminal state $s$ and all previous states within $\mathcal{S}^{\approx}$.*

Thus, for a dynamic construction of the approximation set $\mathcal{S}^{\approx}$, a bound on approximation error can be maintained by tracking the weights of all states in the neighborhood of that set.

In practice, even computing the exact softened distance function for paths through a small subset of states may be computationally impractical. Theorem 3 establishes the approximate search bounds when only a subset of paths in the approximation set are employed to compute the soft distance.

**Theorem 3** *If a subset of paths $\Xi'_{\mathcal{S}_{\approx}} \subseteq \Xi_{\mathcal{S}_{\approx}}$ (and $\bar{\Xi}'_{\mathcal{S}_{\approx}} \subseteq \Xi_{\mathcal{S}_{\approx}} - \Xi'_{\mathcal{S}_{\approx}}$ represents a set of paths that are prefixes for all of the remaining paths within $\mathcal{S}_{\approx}$) through the approximation set $\mathcal{S}_{\approx}$ is employed to compute the soft distance, the error of the resulting estimate is bounded by:*

$$e^{-d_{soft}(s_T)} - e^{-d_{soft}^{\Xi'_{\mathcal{S}_{\approx}}}(s_T)} \leq e^{-\operatorname{softmin}\left(\operatorname{softmin}_{s \in \mathcal{N}(\mathcal{S}_{\approx})}\left\{d_{soft}^{\Xi'_{\mathcal{S}_{\approx}}}(s) + \hat{h}_{soft}(s)\right\}, \operatorname{softmin}_{s \in \mathcal{S}_{\approx}}\left\{d_{soft}^{\bar{\Xi}'_{\mathcal{S}_{\approx}}}(s) + \hat{h}_{soft}(s)\right\}\right)}.$$

### 3.5 Softstar: Greedy forward path exploration and backward cost-to-go estimation

Our algorithm greedily expands nodes by considering the state contributing the most to the approximation bound (Theorem 3). This is accomplished by extending A* search in the following algorithm.

---

**Algorithm 1** Softstar: Greedy forward and approximate backward search with fixed ordering

---

**Input**: State-space graph $\mathcal{G}$, initial state $s_1$, goal $s_T$, heuristic $\hat{h}_{\text{soft}}$, and approximation bound $\epsilon$
**Output**: Approximate soft distance to goal $h_{\text{soft}}^{\mathcal{S}_{\approx}}$
Set $h_{\text{soft}}(s) = d_{\text{soft}}(s) = f_{\text{soft}}(s) = \infty \ \forall \ s \in \mathcal{S}$, $h_{\text{soft}}(s_T) = 0$, $d_{\text{soft}}(s_1) = 0$ and $f_{\text{soft}}(s_1) = \hat{h}_{\text{soft}}(s_1)$
Insert $\langle s_1, f_{\text{soft}}(s_1)\rangle$ into priority queue P and initialize empty stack O
**while** $\operatorname{softmin}_{s \in P}(f_{soft}(s)) + \epsilon \leq d_{soft}(s_T)$ **do**
> Set $s \to$ min element popped from P
> Push $s$ onto $O$
> **for** $s' \in N(s)$ **do**
> > $f_{\text{soft}}(s') = \operatorname{softmin}(f_{\text{soft}}(s'), d_{\text{soft}}(s) + cost(s, s') + \hat{h}_{\text{soft}}(s'))$
> > $d_{\text{soft}}(s') = \operatorname{softmin}(d_{\text{soft}}(s'), d_{\text{soft}}(s) + cost(s, s'))$
> > (Re-)Insert $\langle s', f_{\text{soft}}(s')\rangle$ into P
> **end**
**end**
**while** *O not empty* **do**
> Set $s \to$ top element popped from O
> **for** $s' \in N(s)$ **do**
> > $h_{\text{soft}}(s) = \operatorname{softmin}(h_{\text{soft}}(s), h_{\text{soft}}(s') + cost(s, s'))$
> **end**
**end**
**return** $h_{\text{soft}}$

---

For insertions to the priority queue, if $s'$ already exists in the queue, its estimate is updated to the softmin of its previous estimate and the new insertion estimate. Additionally, the softmin of all of the estimates of elements on the queue can be dynamically updated as elements are added and removed.

The queue contains some states that have never been explored and some that have. The former correspond to the neighbors of the approximation state set and the latter correspond to the search approximation error within the approximation state set (Theorem 3). The softmin over all elements of the priority queue thus provides a bound on the approximation error of the returned distance measure. The exploration order, $O$, is a stack containing the order that each state is explored/expanded.

A loop through the reverse of the node exploration ordering (stack O) generated by the forward search computes complementary backward cost-to-go values, $h_{\text{soft}}$. The expected number of occur-

rences of state transitions can then be calculated for the approximate distribution (5). The bound on the difference between the expected path cost of this approximate distribution and the actual distribution over the entire state set is established in Theorem 4.

**Theorem 4** *The cost expectation inaccuracy introduced by employing state set $\mathcal{S}_\approx$ is bounded by*

$$\left| \mathbb{E}[cost_\theta(\mathbf{S}_{1:T})] - \mathbb{E}_{\mathcal{S}_\approx}[cost_\theta(\mathbf{S}_{1:T})] \right| \leq e^{d_{soft}^{S_\approx}(s_T) - \operatorname*{softmin}_{s \in P}(f_{soft}(s))} \left| \mathbb{E}_P[cost_\theta(\mathbf{S}_{1:T})] - \mathbb{E}_{\mathcal{S}_\approx}[cost_\theta(\mathbf{S}_{1:T})] \right|,$$

*where: $\mathbb{E}_{\mathcal{S}_\approx}$ is the expectation under the approximate state set produced by the algorithm; $\operatorname*{softmin}_{s \in P}(f_{soft}(s))$ is the softmin of $f_{soft}$ for all the states remaining on the priority queue after the first while loop of Algorithm 1; and $E_P$ is the expectation over all paths not considered in the second while loop (i.e., remaining on the queue). $E_P$ is unknown, but can be bounded using Theorem 1.*

### 3.6 Completeness guarantee

The notion of monotonicity extends to the probabilistic setting, guaranteeing that the expansion of a state provides no looser bounds than the unexpanded state (Definition 1).

**Definition 1** *A heuristic function $\hat{h}_{soft}$ is monotonic if and only if $\forall s \in \mathcal{S}, \hat{h}_{soft}(s) \geq \operatorname*{softmin}_{s' \in \mathcal{N}(s)} \left\{ \hat{h}_{soft}(s') + cost(s, s') \right\}$.*

Assuming this, the completeness of the proposed algorithm can be established (Theorem 5).

**Theorem 5** *For monotonic heuristic functions and finite softmin distances, convergence to any level of softmin approximation is guaranteed by Algorithm 1.*

## 4 Experimental Validation

We demonstrate the effectiveness of our approach on datasets for Latin character construction using sequences of pen strokes and ball-handling decisions of professional soccer players. In both cases we learn the parameters of a state-action cost function that motivates the behavior in the demonstrated data and using the softstar algorithm to estimate the state-action feature distributions needed to update the parameters of the cost function [23]. We refer to the appendix for more information.

We focus our experimental results on estimating state-action feature distributions through large state spaces for inverse optimal control as there is a lot of room for improvement over standard approaches which typically use sampling based methods to estimate the distributions providing few (if any) approximation guarantees. Softstar directly estimates this distribution with bounded approximation error allowing for a more accurate estimation and more informed parameter updates.

### 4.1 Comparison approaches

We compare our approach to heuristic guided maximum entropy sampling [15], approximate maximum entropy sampling [12], reversible jump Markov chain Monte Carlo (MCMC) [10], and a search that is not guided by heuristics (comparable to Dijkstra's algorithm for planning). For consistency, we use the softmin distance to generate the values of each state in MCMC. Results were collected on an Intel i7-3720QM CPU at 2.60GHz.

### 4.2 Character drawing

We apply our approach to the task of predicting the sequential pen strokes used to draw characters from the Latin alphabet. The task is to learn the behavior of how a person draws a character given some nodal skeleton. Despite the apparent simplicity, applying standard IOC methods is challenging due to the large planning graph corresponding to a fine-grained representation of the task. We demonstrate the effectiveness of our method against other commonly employed techniques.

**Demonstrated data:**  The data consists of a randomly separated training set of 400 drawn characters, each with a unique demonstrated trajectory, and a separate test set of 52 examples where the handwritten characters are converted into skeletons of nodes within a unit character frame [14].

For example, the character in Figure 1 was drawn using two strokes, red and green respectively. The numbering indicates the start of each stroke.

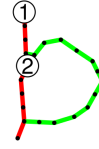

**State and feature representation:** The state consists of a two node history (previous and current node) and a bitmap signifying which edges are covered/uncovered. The state space size is $2^{|E|}(|V| + 1)^2$ with $|E|$ edges and $|V|$ nodes. The number of nodes is increased by one to account for the initial state. For example, a character with 16 nodes and 15 edges with has a corresponding state space of about 9.47 million states.

Figure 1: Character skeleton with two pen strokes.

The initial state has no nodal history and a bitmap with all uncovered edges. The goal state will have a two node history as defined above, and a fully set bitmap representing all edges as covered. Any transition between nodes is allowed, with transitions between neighbors defined as edge draws and all others as pen lifts. The appendix provides additional details on the feature representation.

**Heuristic:** We consider a heuristic function that combines the (soft) minimum costs of covering each remaining uncovered edge in a character assuming all moves that do not cross an uncovered edge have zero cost. Formally, it is expressed using the set of uncovered edges, $E_u$, and the set of all possible costs of traversing edge $i$, $cost(e_i)$, as $\hat{h}_{\text{soft}}(s) = \sum_{e_i \in E_u} \text{softmin}_{e_i} cost(e_i)$.

### 4.3 Professional Soccer

In addition, we apply our approach to the task of modeling the discrete spatial decision process of the ball-handler for single possession open plays in professional soccer. As in the character drawing task, we demonstrate the effectiveness of our approach against other commonly employed techniques.

**Demonstrated data:** Tracking information from 93 games consisting of player locations and time steps of significant events/actions were pre-processed into sets of sequential actions in single possessions. Each possession may include multiple different team-mates handling the ball at different times resulting in a team decision process on actions rather than single player actions/decisions.

Discretizing the soccer field into cells leads to a very large decision process when considering actions to each cell at each step. We increase generalization by reformatting the field coordinates so that the origin lies in the center of the team's goal and all playing fields are normalized to 105m by 68m and discretized into 5x4m cells. Formatting the field coordinates based on the distances from the goal of the team in possession doubles the amount of training data for similar coordinates. The positive and negative half planes of the y axis capture which side of the goal the ball is located on.

We train a spatial decision model on 92 of the games and evaluate the learned ball trajectories on a single test game. The data contains 20,337 training possession sequences and 230 test sequences.

**State and feature representation:** The state consists of a two action history where an action is designated as a type-cell tuple where the type is the action (pass, shot, clear, dribble, or cross) and the cell is the destination cell with the most recent action containing the ball's current location. There are 1433 possible actions at each step in a trajectory resulting in about 2.05 million possible states.

There are 28 Euclidean features for each action type and 29 that apply to all action types resulting in 168 total features. We use the same features as the character drawing model and include a different set of features for each action type to learn unique action based cost functions.

**Heuristic:** We use the softmin cost over all possible actions from the current state as a heuristic. It is admissible if the next state is assumed to always be the goal: $\hat{h}_{\text{soft}}(s) = \text{softmin}_{s' \in \mathcal{N}(s)} \{cost(s, s')\}$.

### 4.4 Comparison of learning efficiency

We compare *Softstar* to other inference procedures for large scale IOC and measure the average test set log-loss, equivalent to the difference between the cost of the demonstrated path, $cost(s_{1:T})$, and the softmin distance to the goal, $d_{\text{soft}}(goal)$, $-\log P(path) = cost(s_{1:T}) - d_{\text{soft}}(goal)$.

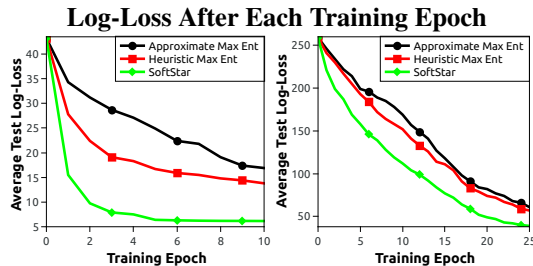

Figure 2: Training efficiency on the Character (left) and Soccer domains (right).

Figure 2 shows the decrease of the test set log-loss after each training epoch. The proposed method learns the models far more efficiently than both approximate max ent IOC [12] and heuristic guided sampling [15]. This is likely due to the more accurate estimation of the feature expectations that results from searching the graph rather than sampling trajectories.

The improved efficiency of the proposed method is also evident if we analyze the respective time taken to train each model. *Softstar* took ~5 hours to train 10 epochs for the character model and ~12 hours to train 25 epochs for the soccer model. To compare, heuristic sampling took ~9 hours for the character model and ~17 hours for the soccer model, and approximate max ent took ~10 hours for the character model and ~20 hours for the soccer model.

### 4.5 Analysis of inference efficiency

In addition to evaluating learning efficiency, we compare the average time efficiency for generating lower bounds on the estimated softmin distance to the goal for each model in Figure 3.

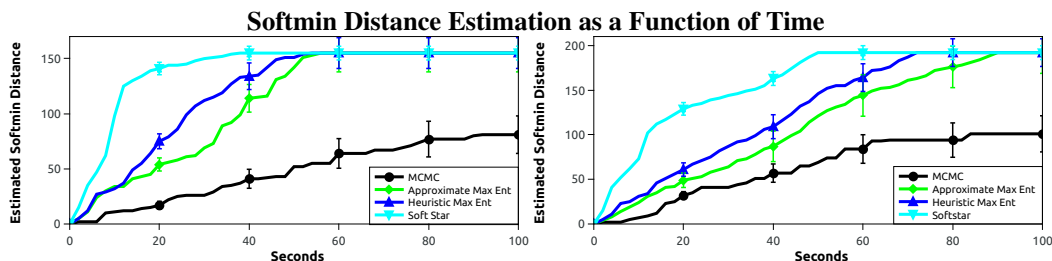

Figure 3: Inference efficiency evaluations for the Character (left) and Soccer domains (right).

The MCMC approach has trouble with local optima. While the unguided algorithm does not experience this problem, it instead explores a large number of improbable paths to the goal. The proposed method avoids low probability paths and converges much faster than the comparison methods. MCMC fails to converge on both examples even after 1,200 seconds, matching past experience with the character data where MCMC proved incapable of efficient inference.

## 5 Conclusions

In this work, we extended heuristic-guided search techniques for optimal planning to the predictive inverse optimal control setting. Probabilistic search in these settings is significantly more computationally demanding than A* search, both in theory and practice, primarily due to key differences between the $\min$ and $\mathrm{softmin}$ functions. However, despite this, we found significant performance improvements compared to other IOC inference methods by employing heuristic-guided search ideas.

## Acknowledgements

This material is based upon work supported by the National Science Foundation under Grant No. #1227495, *Purposeful Prediction: Co-robot Interaction via Understanding Intent and Goals*.

## Footnotes

[1]Equivalently, $\underset{x}{\min}\, \alpha(x) + \underset{x}{\text{softmin}} \left\{ \alpha(x) - \underset{x}{\min}\, \alpha(x) \right\}$ is employed to avoid overflow/underflow in practice.

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
