[Supplementary Material · appendix.pdf]

## A    Proofs

**Proof 1 (of Theorem 1)** *By writing the definition of the entropy, we have:*

$$H(\mathbf{S}_{1:T}) = \mathbb{E}\left[-\log P(\mathbf{S}_{1:T})\right] = \mathbb{E}\left[-\log \frac{e^{-cost_\theta(\mathbf{S}_{1:T})}}{\sum_{\mathbf{s}_{1:T}} e^{-cost_\theta(\mathbf{s}_{1:T})}}\right]$$

$$= \mathbb{E}\left[cost_\theta(\mathbf{S}_{1:T})\right] - d_{soft}(s_{goal}).$$

**Proof 2 (of Lemma 1)** *By definition,*

$$\sum_{path\in\Xi} e^{-cost(path)} = \sum_{path\in\Xi_a} e^{-cost(path)} + \sum_{path\in\Xi_b} e^{-cost(path)}$$

$$\leq \sum_{path\in\Xi_a} e^{-cost(path)} + \sum_{path\in\Xi_b} e^{-\hat{cost}(path)}$$

*for $\hat{cost}(path) \leq cost(path)$. Equivalently, for $\hat{d}_{soft}(s) \leq d_{soft}(s)$: $e^{-d_{soft}^{\Xi}(s)} - e^{-d_{soft}^{\Xi_a}(s)} \leq e^{-\hat{d}_{soft}^{\Xi_b}(s)}$.*

**Proof 3 (of Theorem 2)** *Theorem 2 follows directly from Lemma 1 by choosing the set of paths in the approximation set for $\Xi_a$ and the set of paths not contained within the approximation set for $\Xi_b$. The costs of $\Xi_b$ are lower bounds estimated by adding the cost within the approximation set to the heuristic found at the path's first state outside of the approximation set.*

**Proof 4 (of Theorem 3)** *Following directly from Lemma 1, the paths can be partitioned into the subset $\Xi'_{\mathcal{S}_\approx}$ and the subset of all other paths. The set of all other paths either terminate at $\mathcal{S}_\approx$'s neighbour set or have prefixes represented by the set $\bar{\Xi}'_{\mathcal{S}_\approx}$.*

**Proof 5 (of Theorem 4)** *We will prove by contradiction. Assume that the algorithm does not terminate with guaranteed approximation. There are three cases to consider:*
*1. The algorithm terminates by satisfying the stopping condition. This case violates the stopping condition's approximation guarantee.*
*2. The algorithm terminates by exhausting the set of reachable states. By definition, the algorithm has considered all paths and the estimated softmin distance function is exact.*
*3. The algorithm does not terminate. This case could arise if the algorithm encounters a loop in which no "progress" towards correct approximation is made. We can define this progress primarily by the softmin of the priority queue estimates. For monotonic softmin heuristic functions, expansion of nodes is guaranteed to not increase the priority queue softmin. Thus, an infinite number of states must be expanded for which the monotonic inequality is tight. This violates either the finite softmin assumption or the bounded entropy assumption.*

**Proof 6 (of Theorem 5)** *By definition:*

$$\mathbb{E}\left[cost_\theta(\mathbf{S}_{1:T})\right] = \frac{\sum_{path\in\mathcal{S}_\approx} e^{-cost_\theta(path)}}{\sum_{path\in\mathcal{S}} e^{-cost_\theta(path)}} \mathbb{E}_{\mathcal{S}_\approx}\left[cost_\theta(\mathbf{S}_{1:T})\right]$$

$$+ \frac{\sum_{path\in P} e^{-cost_\theta(path)}}{\sum_{path\in\mathcal{S}} e^{-cost_\theta(path)}} \mathbb{E}_P\left[cost_\theta(\mathbf{S}_{1:T})\right];$$

$$= e^{d_{soft}(s_{goal}) - d_{soft}^{\mathcal{S}_\approx}(s_{goal})} \mathbb{E}_{\mathcal{S}_\approx}\left[cost_\theta(\mathbf{S}_{1:T})\right]$$

$$+ e^{d_{soft}(s_{goal}) - \text{softmin}(P)} \mathbb{E}_P\left[cost_\theta(\mathbf{S}_{1:T})\right].$$

*Then: $\mathbb{E}\left[cost_\theta(\mathbf{S}_{1:T})\right] - \mathbb{E}_{\mathcal{S}_\approx}\left[cost_\theta(\mathbf{S}_{1:T})\right] =$*

$$e^{d_{soft}(s_{goal}) - \text{softmin}(P)}\left(\mathbb{E}_P\left[cost_\theta(\mathbf{S}_{1:T})\right] - \mathbb{E}_{\mathcal{S}_\approx}\left[cost_\theta(\mathbf{S}_{1:T})\right]\right).$$

*Replacing $e^{d_{soft}(s_{goal})}$ with $e^{d_{soft}^{\mathcal{S}_\approx}(s_{goal})}$ increases the positive scalar multiplier preceding the expectation difference, yielding our result.*

## B    Synthetic Evaluation

In our synthetic experiment, we generate a random full cost matrix with 20,000 states. We compute exact soft distance values via Equation 1(6), requiring over 10 minutes for the inversion of the

matrix and evaluate our approach using heuristic functions, $\hat{h}_{\mathrm{soft}}(s) = \alpha h_{\mathrm{soft}}(s)$, with varying values of $\alpha \in [0, 1]$.

Figure 4: Inference efficiency comparison using ratios of the softmin distance as heuristic values.

Figure 4 shows the efficiency of our approach with heuristic functions of varying tightness. Without a heuristic ($\alpha = 0\%$), over 26 minutes of CPU time is required for the proposed algorithm to complete, whereas for $\alpha \geq 75\%$ the softstar algorithm concludes in about 1 second. MCMC performs poorly, getting caught in local optima and taking over 45 minutes of CPU time to converge.

## C   Evaluation Details

### C.1   Feature expectations and gradient computation

In order to generate the desired path distribution we must first run the softstar algorithm in order to generate the softened cost-to-go values, $d_{\mathrm{soft}}$, on the ordered set, $O$. We then use Equation 5 to compute the probability of each state transition in $O$ as well as the expected feature distributions needed to generate the gradient:

$$\nabla L(\theta) = \mathbb{E}_{\hat{\pi}} \left[ \sum_{t=0}^{\hat{T}-1} f(\mathbf{s}_t, \mathbf{s}_{t+1}) \right] - \mathbb{E}_{\tilde{\pi}} \left[ \sum_{t=0}^{\tilde{T}-1} f(\mathbf{s}_t, \mathbf{s}_{t+1}) \right]. \tag{7}$$

We employ stochastic accelerated gradient descent with an adagrad learning rate and accelerated momentum [10, 23] to estimate the cost parameters, $\theta$, using the gradient computation in Equation 7. We update $\theta$ by optimizing the loss function via Equation 7 and matching the estimated distribution of the state-action features to the distribution in the demonstrated trajectories. This is the same method used to update the cost function parameters in previous work [26, 18].

### C.2   Character Drawing

**Feature Representation**   The features are separated into three groups:

- **Four initial transition features:** Destination node distance from each side of the image plane.
- **Nine edge draw features:** Angles of the edge being drawn, with respect to the previous transition angle, to the horizontal, to the vertical, the inverse of these 4 features, and a constant edge draw feature.
- **15 pen lift features:** Each of the edge draw features and initial transition features as well as the distance of the lift and the inverse of the distance of the lift.

An accurate model trained on these features should be informative as to the preferences of how the handwritten characters are drawn in the training set.

**Estimated parameters**   The learned weights indicate that the preferred starting position for a character is in the top left of the frame, drawing an edge is more desirable than a pen lift and that a smoother transition angle is preferred when drawing an edge than executing a pen lift matching previous empirical results on drawing preferences [11].

### C.3 Professional Soccer

**Feature Representation**    There are 28 Euclidean features for each action type and 29 that apply to all action types resulting in 168 total features that form the cost of a state transition between the current cell, the previous cell, and the destination cell. We use the same features as the character drawing model and include a different set of features for each action type in order to learn unique action based cost functions that better represent the individual characteristics of each action type. We also include a set of 29 features that apply to all types and are used to learn common characteristics between actions and speed learning for a total of 174 features.

**Estimated parameters**    The learned weights indicate that the players prefer close range shots in front of the goal, crosses from the sides of the field, and players tend to take actions that move the ball closer to the goal in short distances. A more detailed analysis of learned behavior will be reserved for future work.