[Reviews · NeurIPS 2015]

Submitted by Assigned_Reviewer_1

I think this paper presents a valuable contribution to inverse optimal control and more generally to analysis of decision making and control as probabilistic inference. Incorporating ideas from heuristic search to estimate state marginals under the maxent model is an interesting and thought-provoking idea. The authors also evaluate their method on two interesting real-world datasets, including a football prediction dataset.

I think that there is a lot to learn from this work, and I can see some follow-up working making good use of the ideas presented in this paper.

That said, I think the paper is seriously hampered by confusing writing and excessive "specialization". For example, it took me several passes through the paper to understand that the authors were actually estimating the model parameters theta, which is mentioned briefly in the experiments section and at the end of Sec 2.2. Most of the paper is concerned with inference, which is indeed the most pressing problem in the maxent IOC framework, but the authors don't really explain this very well in the background section. The main technical sections of the paper are quite heavy on derivation and light on intuition and motivation, which also makes them tough to follow. Overall, I think the paper can be improved dramatically from a more thorough top-down organization that clearly lays out what the authors are trying to do, better explains the maxent model, and clarifies where the proposed component fits in in the context of a practical algorithm.
Summary: The authors present a very interesting method for estimating marginals under the maximum entropy IOC model based on heuristic search (inspired by A*). The method is evaluated on two realistic tasks and shows considerable promise, but the paper is hampered somewhat by unclear writing.

Submitted by Assigned_Reviewer_2

In this paper, the authors extend the heuristic search algorithm A*, to a softened version.

Here, you do not search for a single optimal path, but one based (rougly) on maximizing an expected cost, w.r.t. a distribution over paths.

The authors propose in particular an approximation algorith, with some guaratnees.

The paper is motivated in terms of inverse optimal control (inverse reinforcement learning), although this domain lies outside my expertise, and I did not judge the relevance of this problem in this domain.
Summary: The idea of a softened A* search is conceptually appealing, and the authors appear to have done a reasonable job at formulating the problem, and proposing an approach to tackling it.

Submitted by Assigned_Reviewer_3

The paper generalizes A* search to maxent inverse reinforcement learning (for problems with large search space), which is shown to be both faster and better than existing approximation approaches.

The proposed method is clearly written. However, there's little description for the baselines/related work. How is this different from prior works? Does the improvement come from softmax or A* search?

The results shows the average difference between costs of the demonstrated trajectories and the softmin path. This is a sensible choice. I'd expect there is a task loss based on the demonstration and the path found by the learned policy though. How is \epsilon chosen for the experiments?

Minor: Algorithms 1 does not involve updates for \theta. How is it learned? f_{soft}(.,.) is not defined.

Summary: This paper proposes an A*-like heuristic search method for maxent inverse reinforcement learning, and provides theoretical guarantees. The experimental results look promising. It'll be nice if there is some explanation for why it works better than the baselines.

Submitted by Assigned_Reviewer_4

This paper extends heuristics from the A* algorithm to the regime of inverse optimal control, for the purpose of inferring behavioural intents from data.

The paper considers a probability distribution over possible paths, and efficient approximations to this distribution with a small subset of possible paths.

The method is tested on writing Latin characters, and footage of professional soccer.

A significant weakness of this paper is that it does not clearly define the problem being solved.

Although two concrete experiments are described, I could not clearly identify the task from the experiment descriptions.

For example on the character drawing task, what is predicted and when? The pieces of the solution seem locally consistent and potentially interesting as ideas (extending heuristics to the probabilistic planning setting where one must consider a collection of possible paths), but I could not see where the overall task was clearly defined.

Summary: This paper extends heuristics from the A* algorithm to the regime of inverse optimal control, for the purpose of efficiently inferring behavioural intents from data.

Unfortunately, I could not determine what problem this paper was formally trying to solve.

Submitted by Assigned_Reviewer_5

Authors apply efficient heuristic-guided search algorithms for optimal planning to inverse reinforcement learning based on A* algorithm. It is only applicable in discrete sate-space graph.

It is clear and reads well.

Quality: I found the quality of research discussed in the paper to be fine.

Clarity: For the most part the paper is clearly written.

Originality: The paper seems original.

Significance: I am not sure if this line of work will have significant impact as it is only applicable in discrete sate-space graph.

Seemingly small example is still very hard t o solve.

I suggest authors to look into natural language processing application, such as Ross et al 2011 for more realistic application.

Stephane Ross, Geoffrey J. Gordon, and Drew Bagnell. A reduction of imitation learning and structured prediction to no-regret online learning. In 14th International Conference on Artificial Intelligence and Statistics, pages 627-635, 2011.
Summary: It is clear and reads well.

I found the quality of research discussed in the paper to be fine.

Author Feedback
Author rebuttal: We appreciate all of the suggestions given in the reviews and will use the comments to improve the next version of the paper and clarify any issues brought up by the reviewers.

The problem being addressed is the inefficiency of existing inference methods for reward-based probabilistic models of behavior (R5). We develop a method using ideas from heuristic-based search (i.e., A*) for estimating path distributions through large scale deterministic graphs with bounded approximation guarantees (R5). This is an improvement over previous methods as it results in more accurate distribution estimations without the complexity/sub-optimality concerns of path sampling (R2). This approach is suitable for any problem that can be represented as such a graph.

Additionally, since softstar does not sample paths, but instead searches the space as in A*, it does not need to retrace its steps along a previously searched trajectory to find a new path to the goal. It will instead create a new branch from an already explored state. Sampling would require retracing an entire sequence until this branching state was reached. This allows for improvements in efficiency in addition to the distribution estimation improvements.

The ideas of our paper can be extended to settings with non-deterministic dynamics, but not trivially. We believe this would make an already complex idea lying at the intersection of probabilistic graphical models and search-based planning even more difficult to explain. Thus, we reserve generalization to stochastic settings for future work (e.g., a journal version of this work) (R3) (R6).

We focus our experimental results on estimating state-action feature distributions through large state spaces as there is a lot of room for improvement over the standard approaches used in inverse optimal control where sampling based methods are typically used to estimate the path distribution through the space which provide few (if any) approximation guarantees (R2) (R5). Softstar directly estimates this distribution with bounded approximation error allowing for a more accurate estimation which leads to more informed parameter updates for inverse optimal control.

We left the details of the parameter updates for the cost function in the appendix as we did not make any contributions to how the parameters are learned once a path distribution is estimated and are instead considering how a more accurate distribution affects learning in inverse optimal control (R7). \theta is updated by optimizing the loss function in equation 10 in the appendix (R2). This matches the estimated distribution of the state-action features using the learned parameters to the distribution in the demonstrated trajectories. We will make this more clear earlier in the paper.

The character drawing task is to learn the behavior of how a person draws a latin character given some nodal skeleton of the character and the professional soccer task was in estimating the decision process of the team in possession of the ball (R5). In both cases we are learning the parameters for a state-action feature based cost function that motivates the behavior in the demonstrated data. The log-loss between the distribution found using the learned parameters and the distribution of the demonstrated paths is represented in figure 2 which shows how this loss changes throughout training as the model learns from the demonstrated behavior (R2).

We were unfortunately pressed for space and referred to the cited papers for information on the baselines used. We will add more information on the baselines in the appendix (R2).

Epsilon is chosen to be log(10) for these experiments where the size is dependent on the approximation error that has been decided should be allowed (R2).